# Test-Time Scaling for Multistep Reasoning in Small Language Models via A* Search

## Abstract

Large language models (LLMs) have demonstrated strong abilities across various tasks but are costly in computation and memory. In contrast, Small Language Models (SLMs) offer significant advantages in efficiency and deployability but usually struggle with complex mathematical reasoning tasks. To tackle this issue, we present the Test-time A* Search (TTA*), a test-time scaling framework that casts reasoning as a goal-directed search over a tree of partial solutions, guided by an A*-style cost function. TTA* is training-free and requires no external supervision or multi-model structure, making it practical in resource-constrained settings. As a drop-in decoding wrapper for SLMs, TTA* systematically explores, critiques, and refines candidate solution paths via its own self-reflection capability. Extensive experiments on popular mathematical reasoning benchmarks and a variety of base models show that TTA* consistently improves accuracy and robustness, indicating broad applicability to general mathematical reasoning tasks.

## 1 Introduction

Large Language Models (LLMs) such as GPT-5 OpenAI (2025) have demonstrated impressive capabilities across a wide range of AI tasks. However, their high performance comes at the cost of substantial computational and memory requirements, which complicates personalization and limits deployment in resource-constrained settings. Small Language Models (SLMs), in contrast, offer an attractive alternative due to their efficiency, lower latency, and ability to run locally—properties that are especially valuable in remote or resource-limited environments Cheng et al. (2024); Wang et al. (2024); Al-Garadi et al. (2025). Because of their adaptability and practicality, SLMs are increasingly viewed as a promising avenue for agentic AI Belcak et al. (2025). Nevertheless, they still face challenges in complex reasoning tasks, particularly in high-stakes domains such as mathematics and healthcare Ma et al. (2025); Ahn et al. (2024). Specifically, SLMs often underperform on multi-step reasoning and remain prone to hallucination—issues that are critical for both safety and reliability Huang et al. (2025); Zhou et al. (2025); Glazer et al. (2024); Lewkowycz et al. (2022).

To address these challenges, a growing body of work seeks to enhance the reasoning capability of small models. Many approaches rely on additional training, for example with preference data from human feedback or guidance from larger teacher models, to instill better stepwise reasoning Ong et al. (2025). To avoid the cost and complexity of retraining, *test-time scaling* methods have been proposed, in which the model allocates extra inference-time computation—often via structured search—to improve its answers. However, most existing test-time scaling techniques depend on ancillary components such as progress reward models (PRMs) or other external guidance, which undermines deployability and increases engineering burden. A complementary line of work explores self-rewarding or self-reflection strategies, where the model critiques its own intermediate outputs. While promising, these methods can compound errors over multiple steps due to the limited evaluative capacity of small models.

In this paper, we pursue a practical *test-time scaling* approach with self-reflection for small language models, aiming to boost reliability without additional, infeasible training costs Xia et al. (2024). We introduce **Test-time A* Search (TTA*)**, a framework that integrates heuristic search with SLMs to improve solution quality on complex reasoning tasks. TTA* casts reasoning as a goal-directed search over a tree of partial or imperfect solutions: nodes encode candidate derivations, edges apply iterative refinements, and expansions are prioritized by an A*-style score Hart et al. (1968) that blends a cost function from the length of the search path (cost-to-come) with a heuristic of future potential from

model's self-evaluation (cost-to-go). The procedure is *training-free*, requires no external supervision or multi-model orchestration, and supports explicit compute budgets with anytime behavior—making it well suited for resource-constrained deployments. As illustrated in Figure 1, our contributions are as follows:

- We formulate multi-step reasoning as a guided tree search and leverage the model's self-reflection to define the heuristic, prioritizing expansions toward high-quality solutions.

- To mitigate hallucinations and compounding errors, we design an A*-style cost function that balances exploration with skepticism about self-assessments, avoiding overuse of self-reflection while improving efficiency under fixed compute budgets.

- We introduce a calibration-aware scoring mechanism that enables consistent evaluation of partial and final answers, supporting robust multi-step reasoning.

- We conduct comprehensive experiments across a variety of benchmarks and SLM backbones. Results show that TTA* improves accuracy and matches the reasoning performance of larger, more expensive models without requiring additional training, external supervision, progress-reward models, or multi-model orchestration.

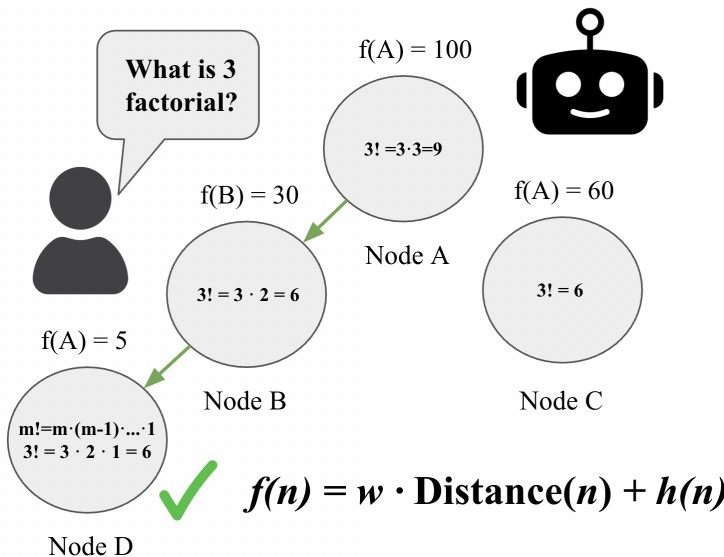

Figure 1: An example of the TTA* method for the given question. Green edges represent the chosen reasoning path. Each node is a candidate answer, and the tree structure shows how answers improve through progressive exploration.

## 2 RELATED WORK

### 2.1 MULTI-STEP REASONING IN LLMS

Recent advances in LLM research highlight their ability to perform multi-step reasoning. Chain-of-thought (CoT) prompting Wei et al. (2023) exposes intermediate steps, improving performance on complex tasks such as mathematical problem solving. Beyond single-path CoT, search-based methods explore multiple reasoning trajectories before producing a final answer. In particular, tree search techniques integrate with LLMs to expand and prune reasoning branches: Monte Carlo Tree Search (MCTS) has been applied to explore intermediate steps, evaluate partial solutions, and backtrack from low-value branches Xie et al. (2024); Zhang et al. (2024a). These approaches systematically balance exploration and exploitation, allowing models to consider alternative reasoning paths and recover from early errors. However, many of these methods rely on external reward models, curated supervision (e.g., outcome-based or process-based rewards), or fine-tuned evaluators. Such dependencies can limit practicality in constrained deployment settings or when access to large models is restricted.

Earlier work also explored distilling reasoning capabilities from large LLMs into smaller models Zhu et al. (2024); Zhang et al. (2025). While distillation can transfer some reasoning skills, it is computa-

tionally intensive, often requires teacher models with high accuracy ceilings, and provides limited gains when task distributions shift. Other studies have leveraged stronger models to generate synthetic datasets for fine-tuning smaller models Liu et al. (2023), or used larger models as test-time supervisors Yang et al. (2025b), but both approaches introduce additional compute burdens and dependency on external resources, restricting applicability in low-resource or test-time-only scenarios.

## 2.2 SMALL LANGUAGE MODELS

Small Language Models (SLMs) are particularly attractive for deployment in resource-constrained environments, including edge devices and mobile platforms, due to their low memory footprint and computational requirements Belcak et al. (2025). Enhancing their reasoning capabilities allows these lightweight models to handle complex tasks efficiently, without relying on large-scale LLMs, enabling real-time inference and practical applications in low-power or embedded settings.

Improving mathematical reasoning in SLMs has been approached via knowledge distillation, fine-tuning, and inference-time assistance from larger models. Knowledge distillation Zhu et al. (2024); Zhang et al. (2025) transfers reasoning knowledge from a high-performing teacher model to an SLM, but is computationally expensive, time-consuming, and bounded by the teacher's capabilities. Inference-time supervision, such as *Speculative Thinking*, pairs a small model with a stronger "guide" to handle difficult reflective steps Yang et al. (2025b). While this can improve performance, it introduces dependency on real-time access to a large model.

Other methods, like *rStar-Math*, combine SLMs with internal reward models to drive search and iterative self-evaluation Guan et al. (2025). These frameworks are effective, but require multi-round data generation, reward model training, and careful orchestration between models, increasing engineering complexity and compute costs. Direct fine-tuning methods, including Solution Guidance Bi et al. (2025) or reinforcement learning-based approaches Dang & Ngo (2025), similarly incur high costs, rely on additional supervision, and limit accessibility for low-resource settings.

## 2.3 SELF-REFLECTION AND SELF-REWARDING

An alternative line of work equips LLMs with *self-evaluation* signals to guide reasoning. Here, the model generates candidate derivations and attaches lightweight diagnostics, such as confidence estimates, critiques, or partial checks, which inform revisions, backtracking, or reordering. Such mechanisms reduce ungrounded derivations and help prioritize promising reasoning branches without relying on ground-truth labels. Recent tree-search systems incorporate internal assessments as *progress heuristics* or *value estimates* to steer exploration, sometimes augmented with verifier-like signals or learned progress models Xie et al. (2024); Zhang et al. (2024a).

Although self-guided evaluations remove dependence on external supervision, they are prone to error amplification when the evaluator is weak, which is common in small models Zhu et al. (2024). Consequently, small models' self-evaluation can be noisy, leading to overconfidence in incorrect answers or excessive critique in accurate responses Zhang et al. (2024b). RLHF-style self-critique Lee et al. (2024) has been studied to address this issue, but typically requires additional training, external supervision, or complex multi-step orchestration, substantially increasing the engineering burden—a demand often impractical for SLM deployment.

Our work builds on this direction but explicitly *binds* self-evaluation to an A*-style cost function that balances cost-to-come and cost-to-go. This mitigates overconfidence and limits error propagation during search. Importantly, our self-reflection mechanism functions entirely within this framework without additional training, external supervision, or complex orchestration, making it suitable for real-world SLM deployment.

## 2.4 TRAINING-FREE AND INFERENCE-TIME METHODS FOR SLMS

In contrast to prior approaches, our method is a single-model, training-free, test-time strategy that frames multi-step reasoning as a heuristic search problem. By deriving the heuristic entirely from the model's own critiques and self-evaluations, we remove the need for external rewards, teacher models, or fine-tuning. At the same time, structured exploration and self-correction are preserved. This yields a practical, low-resource framework for improving SLM reasoning that combines tree-based search with model-internal feedback, scales naturally with model size and complexity, and provides a deployable alternative for enhancing small models' reasoning capabilities in real-world scenarios.

## 3 METHODOLOGY

### 3.1 MOTIVATION

LLM reasoning frequently spans long, multi-step derivations Shen et al. (2025). When treated as a single forward pass, early mistakes propagate and compound, degrading reliability in tasks such as mathematical problem solving. This is only exacerbated in SLMs, which lack the same depth as LLMs and are often observed to hallucinate or repeat mistakes Ma et al. (2025).

To improve trustworthiness, reasoning should be decomposed into structured, incremental tasks that build upon one another, allowing the model to maintain coherence and reduce the risk of compounding errors Wu & Tsioutsiouliklis (2024). Logically, long and complex problems—where LLMs are more prone to hallucination—should be broken down into smaller, manageable steps that can be sequentially solved and then combined to reach a final solution.

To effectively manage the complexity above, we propose representing the LLM reasoning process as a tree. This structure naturally supports the exploration of multiple reasoning trajectories, enabling branching, backtracking, and iterative refinement.

Prior research has demonstrated that such approaches can significantly boost accuracy through iterative refinement and the identification of high-quality leaf nodes Zhang et al. (2024a); Ahn et al. (2024); Xie et al. (2024). However, many of these tree search approaches for LLMs rely on auxiliary components—such as reward/value models, progress reward models, or additional fine-tuning—to evaluate partial solutions and guide exploration, which undermines deployability in resource-constrained environments. Our goal is to build on this foundation, leveraging tree-based reasoning to improve reasoning accuracy, specifically in limited-capacity models.

### 3.2 TREE-BASED REASONING

In this subsection, we begin with the formulation of the tree-based reasoning as illustrated in Figure 1.

In particular, each node $n$ represents a partial, intermediate, or imperfect reasoning attempan t and edge denoted by $e(n_1, n_2)$ describes the self-reflection and refinement between node $n_1$ to node $n_2$. By treating each node as an independent candidate solution, we can assign evaluation scores using techniques such as self-consistency Wang et al. (2023), model-generated critiques Yu et al. (2025), or correctness-based metrics. These numerical scores enable direct quantitative comparison between nodes, allowing us to frame multi-step reasoning as a search problem whose objective is to efficiently expand nodes that are both promising and near a complete solution.

As previously mentioned, the self-reflection capabilities of Small Language Models (SLMs) are generally less reliable than those of larger models. Because of this, it is beneficial to incorporate other factors when evaluating nodes, so that the search is guided not only by self-reflection but also by additional metrics or heuristics. These combined signals can then inform a more reasonable search, helping prioritize nodes that are both promising and trustworthy.

### 3.3 A* SEARCH

The A* search is a best-first algorithm that balances the cost to reach a node with a heuristic estimate of the cost to the goal Hart et al. (1968), defined as:

$$n_{\text{next}} = \arg \min_{n \in \mathcal{N}_{\text{visited}}} f(n) = g(n) + h(n), \tag{1}$$

where $\mathcal{N}_{\text{visited}}$ is the collection of *visited* nodes in a tree search task, $g(n)$ is the cost of reaching node $n$, and $h(n)$ is the heuristic function defined by the self-critic / self-reflection of the language models. As suggested in Hart et al. (1968), the path of the A* search is guaranteed to be the shortest path as long as the heuristic function $h(\cdot)$ is *admissible* (i.e., $h(\cdot)$ never overestimates the cost of reaching the goal).

In our setting, $g(n)$ represents the accumulated "cost" of reasoning steps taken to reach the current node. Intuitively, nodes that require many corrections or refinements from previous steps have higher $g(n)$, while nodes closer to a coherent and promising partial solution have lower $g(n)$. This prevents the search from blindly expanding nodes that appear promising according to the heuristic $h(n)$ but are built on long, error-prone chains of reasoning.

We use A* instead of purely greedy search because relying solely on the self-reflection heuristic $h(n)$ can be misleading: small language models (SLMs) may provide noisy or inconsistent evaluations. A* naturally balances the heuristic with the path cost $g(n)$, allowing the algorithm to favor nodes that are both promising and reachable with minimal accumulated error.

## 3.4 ADAPTING A* SEARCH TO TREE-BASED REASONING

We reinterpret the A* search algorithm in the context of LLM-based reasoning by modeling the reasoning process as a search through a structured tree space of potential answers.

To navigate this space efficiently, we define a modified A* cost function that balances exploration and exploitation. We define $g$ and $h$ as:

$$
\begin{aligned}
g(n) &= w \cdot \text{Distance}(n) \\
h(n) &= 100 - \text{Reward}(n) \\
f(n) &= g(n) + h(n)
\end{aligned}
\tag{2}
$$

Distance$(n)$ is the depth of the node from the root, encouraging broad exploration, and $w$ controls the exploration-exploitation tradeoff. Reward$(n)$ is derived from correctness, self-consistency, or model-generated critiques using the same LLM.

While self-reflection in Small Language Models (SLMs) is noisy and unreliable, computing self-consistency across multiple candidate answers helps stabilize the evaluation. By aggregating judgments over several independently generated completions, self-consistency reduces the variance caused by flawed single-step self-reflection, making the heuristic $h(n)$ a more reliable guide in the A* search.

This formulation promotes efficient traversal of the reasoning tree by allowing the algorithm to favor nodes that are either close to the root, encouraging breadth of search, or have strong reward signals, indicating correctness or promise. Crucially, this ensures that the driving force of the search is not solely the model's self-reflection, which—while greedily selecting apparently promising nodes—can be unreliable and noisy in Small Language Models (SLMs). By combining reward signals with exploration through the tree structure and additional heuristics, the search is guided by both promising evaluations and complementary factors that mitigate the risk of over-relying on flawed self-assessments.

In practice, this setup enables the LLM to iteratively refine solutions by generating and evaluating child nodes, then selecting the next node to expand based on the lowest estimated cost $f(n)$.

## 3.5 TREE-BASED REASONING PROCEDURE

We model SLM reasoning as a tree search problem, where nodes represent candidate answers and edges correspond to refinements. The search begins at a root node and iteratively generates new candidates through critique and revision, exploring the most promising paths in the reasoning space. All prompts used are provided in Appendix A.

**Node Structure:** Each node stores the question, candidate answer, critique, and reward score. This reward score is used in the A* cost function $f(n)$ to guide node selection, balancing exploration and exploitation.

**Initialization:** The root node is initialized by prompting the model to solve the problem using the phrase *"Let's think step by step."* We set this root node as the current node, initialize a list to store all visited answers with their corresponding reward scores, and maintain a separate list of nodes yet to be explored.

**Tree Traversal:** We repeat the following process a pre-defined number of times, constrained by $max\_iterations$.

**Critique:** The current node is critiqued by the same model, which provides detailed, constructive feedback on how to improve the answer while also highlighting what has been done correctly.

**Self-Evaluation:** After the critique, each node is assigned a reward score estimating its "distance" from the correct answer, which is then used to compute $h(n)$ in the A* cost function. To generate this score, the LLM is prompted with the original question and the candidate answer to assign a numerical grade from 0 to 100 based on correctness, coherence, and completeness. To reduce variance and

improve reliability, the LLM is queried multiple times (e.g., three independent evaluations), and the resulting scores are averaged. This self-consistent evaluation produces a more stable reward function, enabling more reliable comparisons between candidate answers without requiring external reward models or additional training.

**Tree Expansion:** After incorporating the critique and reward scores, we generate two child nodes by prompting the model to revise the parent node's answer. Each child node represents an attempted refinement or improvement over its parent, explicitly addressing the feedback provided in the parent's critique. Distinct children can be obtained by sampling multiple outputs from the LLM (using temperature sampling) to introduce variation between the refinements. For each child node, we then generate a new critique and self-evaluation score. Both child nodes are added to the list of nodes yet to be explored. The next node to expand is selected using the A* cost function, choosing the node with the lowest $f(n)$. This node becomes the current node and is appended, along with its score, to the list of all generated answers. Two child nodes are generated per iteration for simplicity and to maintain a tractable search. This strikes a balance between exploring multiple candidate solutions and limiting computational cost.

**Final Answer Selection:** The search iterates up to a fixed number of steps ($max_{i}terations$). However, if any candidate achieves a self-evaluation score above a pre-defined threshold (e.g., 95/100), the search terminates early. The final answer is selected as the candidate with the highest self-evaluation score among all generated nodes, ensuring both efficiency and high-quality solutions.

A formal write-up of this algorithm is provided in Algorithm 1, and all our code is available on GitHub.[1]

---

**Algorithm 1** A* for LLMs

---

**Require:** Problem prompt (e.g., "What is 4 times 3?")
**Ensure:** Answer with maximum LLM evaluation score
1: $prompt \leftarrow$ "What is 4 times 3? Let's think step by step."
2: $root\_node \leftarrow$ LLM($prompt$)
3: $AnswerStorage \leftarrow \{\}$          ▷ Stores ($answer, critique, score$) tuples
4: $current\_node \leftarrow root\_node$
5: Generate critique for $current\_node$
6: $score \leftarrow$ LLM evaluation of $current\_node$ (range 0–100)
7: Add ($current\_node, critique, score$) to $AnswerStorage$
8: **for** $i = 1$ **to** $max\_iterations$ **do**
9:    Compute $f(n) = w \cdot \text{Distance}(n) + h(n)$ for all unvisited leaves
10:    $current\_node \leftarrow$ leaf with minimum $f(n)$      ▷ Select node to expand
11:    Generate two children of $current\_node$
12:    **for** each child **do**
13:      Include parent answer and critique in prompt
14:      Generate critique for child
15:      $score \leftarrow$ LLM evaluation of child (range 0–100)
16:      Add ($child, critique, score$) to $AnswerStorage$
17:    **end for**
18: **end for**
19: **return** answer in $AnswerStorage$ with maximum score

---

## 4 EXPERIMENTS

### 4.1 MODELS

As previously mentioned, our work looks to focus on SLMs due to their reduced memory footprint and low inference cost, which make mathematical reasoning available without prohibitive GPU requirements. For example, we prioritize models in the 1–8B parameter range that typically require only 2–16 GB of VRAM, allowing deployment on consumer-grade GPUs or even embedded devices. In contrast, larger 70B models can require 140 GB of GPU memory or more, restricting their use to high-performance computing clusters Subramanian et al. (2025). By prioritizing applicability over

---

[1] `https://anonymous.4open.science/r/TTASearch/README.md`

scale, we aim to highlight the broad usage of LLMs in realistic, resource-constrained environments rather than maximizing performance on specialized hardware.

To align with our focus on wide deployment, we conduct all experiments using open-source models from two major families: LLaMA and Qwen. Specifically, we use LLaMA-3.2-1B Meta AI (2024b), Qwen3-4B (Base) Yang et al. (2025a), LLaMA-3-8B Grattafiori et al. (2024), and LLaMA-3.1-8B Meta AI (2024a) to represent widely adopted general-purpose SLMs. In addition, we include Qwen2.5-Math-7B Yang et al. (2024), a math-specialized model. We choose these models for their open-source availability and ease of deployment, making them representative examples of off-the-shelf SLMs. Together, these models offer a diverse but practical testbed for evaluating our method.

For all models, we set the decoding temperature to 0.3 to encourage mostly deterministic outputs while still allowing limited diversity, following prior work on reasoning evaluations.

### 4.2 DATASETS

We evaluate on four mathematics benchmarks spanning a range of difficulty:

- **GSM8K** Cobbe et al. (2021): Grade-school word problems requiring multi-step arithmetic and commonsense reasoning.

- **MATH500** Lightman et al. (2023): Competition-style problems that require high school-level reasoning and complex logic.

- **AIME (2024)** Mathematical Association of America (2024): Problems from the American Invitational Mathematics Examination, known for their difficulty and requirement of creative algebraic problem-solving.

- **MATH401** Yuan et al. (2023): A collection of 401 problems covering arithmetic and elementary functions (e.g., trigonometry, logarithms) designed to test reliability on fundamental operations.

This collection of datasets allows us to evaluate models on both basic reasoning and advanced problem-solving, while maintaining focus within the mathematical reasoning domain.

### 4.3 EVALUATION

### 4.4 EVALUATION

Table 1 and Figure 2 summarize the performance of our experiments across four benchmarks: GSM8K, MATH500, AIME (2024), and MATH401. Overall, TTA* consistently improves reasoning accuracy across both LLaMA and Qwen model families.

On the **AIME (2024)** benchmark, which is the most challenging, relative improvements are particularly notable. For example, LLaMA-3.1-8B improves from 3.3% to 10.0% (+6.7 points, +203% relative), while Qwen2.5-Math-7B improves from 6.7% to 10.0% (+3.3 points, +49.3%). These results indicate that TTA* effectively enhances structured reasoning even on competition-style problems.

On **MATH500**, TTA* delivers substantial gains for both model families, with relative improvements ranging from +27.6% to +71.0%. This demonstrates that structured search benefits scale with problem complexity, allowing small and mid-sized LLMs to achieve competitive reasoning performance.

On more standard benchmarks like **GSM8K** and **MATH401**, performance gains are consistent and reliable. For GSM8K, relative improvements range from +9.5 to +18.9, while on MATH401 (arithmetic and elementary functions), relative gains range from +16.2 to +26.1. This suggests that TTA* not only improves advanced reasoning but also reinforces fundamental problem-solving skills.

Overall, these results demonstrate that TTA* consistently improves reasoning performance across a wide range of model scales and benchmarks, without requiring additional model training. The improvements suggest that TTA* mitigates general reasoning limitations in SLMs rather than exploiting model-specific idiosyncrasies. Gains are observed across both general-purpose and specialized SLMs, small and mid-sized architectures, and tasks of varying difficulty. All baselines are evaluated against zero-shot chain-of-thought Wei et al. (2023), and full prompts are provided in the appendix.

| Name | GSM8K | MATH500 | AIME (2024) | MATH401 |
|---|---|---|---|---|
| **LLaMA Models** | | | | |
| Llama-3.2-1B | 39.7 | 29.8 | 3.3 | 58.5 |
| w/ TTA* | 51.2 | 39.4 | 6.7 | 68.1 |
| Δ | 11.5 | 9.6 | 3.4 | 9.6 |
| | ↑ 29.0% | ↑ 32.2% | ↑ 103.0% | ↑ 16.4% |
| Llama-3-8B | 75.4 | 26.2 | 3.3 | 62.8 |
| w/ TTA* | 87.1 | 44.2 | 6.7 | 78.3 |
| Δ | +11.7 | +18.0 | +3.4 | +15.5 |
| | ↑15.5% | ↑68.7% | ↑103.0% | ↑24.7% |
| Llama-3.1-8B | 77.2 | 52.2 | 3.3 | 68.8 |
| w/ TTA* | 90.2 | 66.6 | 10.0 | 80.1 |
| Δ | +13.0 | +14.4 | +6.7 | +11.3 |
| | ↑16.8% | ↑27.6% | ↑203.0% | ↑16.4% |
| **Qwen Models** | | | | |
| Qwen3-4B | 84.8 | 53.9 | 23.3 | 62.1 |
| w/ TTA* | 95.9 | 73.4 | 30.0 | 78.3 |
| Δ | 8.3 | 19.5 | 6.7 | 16.2 |
| | ↑9.5% | ↑36.2% | ↑28.8% | ↑26.1% |
| Qwen2.5-Math-7B | 73.3 | 46.8 | 6.7 | 65.5 |
| w/ TTA* | 87.1 | 74.2 | 10.0 | 78.9 |
| Δ | +13.8 | +27.4 | +3.3 | +13.4 |
| | ↑18.9% | ↑58.5% | ↑49.3% | ↑20.5% |

Table 1: Reasoning accuracy for TTA*. Baseline comparison is made with zero-shot CoT Wei et al. (2023). Absolute and relative percentage improvements are presented.

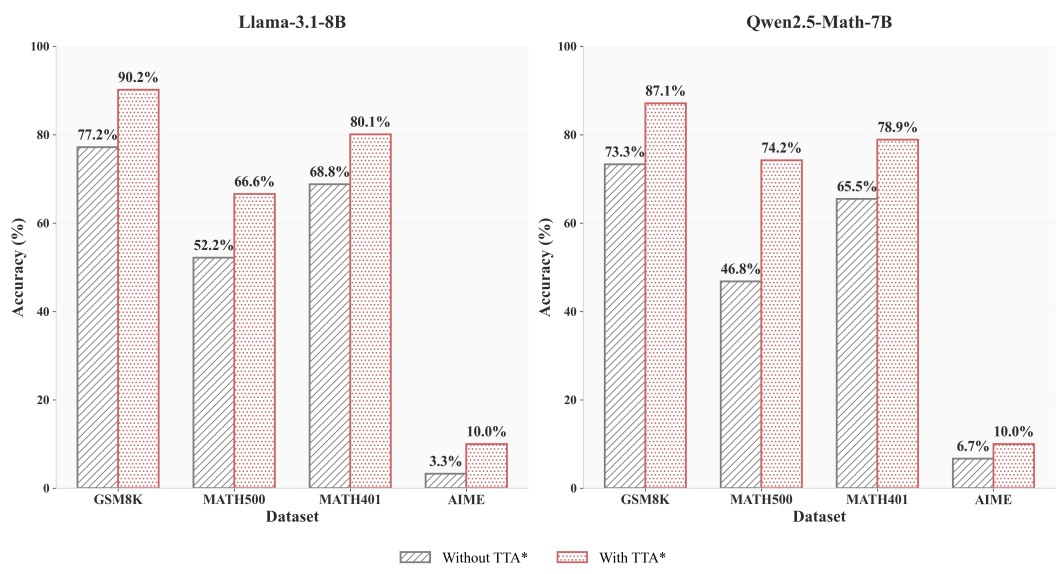

Figure 2: Performance comparison of TTA* methodology across different mathematical reasoning benchmarks for Llama-3.1-8B (left) and Qwen2.5-Math-7B (right).

Table 2: Comparison of model sizes, VRAM requirements, and performance across datasets.

| Model | Parameters | VRAM Required | Accuracy / Score |
|---|---|---|---|
| Llama 3.1 8B | L8B | 16 GB | 90.2% |
| Llama 3.1 70B | 70B | 140 GB | 95.1% |
| GPT-4 | 1.76T | 3.3 TB | 92.0% |

## 5 PRACTICALITY AND RESOURCE CONSTRAINTS

As the adoption of large language models (LLMs) continues to grow across business, education, healthcare, and software development Cheng et al. (2024), practical considerations such as hardware requirements, cost, and deployment feasibility become critical. Many users and organizations lack the financial and technical resources to consistently maintain state-of-the-art hardware capable of supporting powerful LLMs, which often consist of tens or hundreds of billions of parameters. For example, 70B-parameter models typically require more than 100 gigabytes of VRAM, far exceeding the capacity of a single 80–100 GB GPU. In practice, large models must be shared across 4–8 high-end GPUs (e.g., A100/H100s), costing tens of thousands of dollars per GPU, and inference costs per token can be an order of magnitude higher than smaller models. In contrast, smaller models, ranging from 1B to 8B parameters, can run on consumer-grade GPUs (e.g., 8–16 GB VRAM) or even CPUs, dramatically lowering the barrier to entry while maintaining competitive performance in many tasks.

By focusing on 1–8B parameter models, which typically require only 2–16 GB of VRAM, our approach enables deployment on embedded and edge devices, remote servers with limited memory, and consumer-grade hardware.

## 6 CONCLUSION

In this paper, we introduced Test-Time A* Search (TTA*), a novel framework that equips language models with structured, iterative reasoning through tree-based search.

Our work is motivated by the critical issue of providing weaker, yet more accessible models with advanced reasoning abilities. This emphasis is crucial for promoting the eventual usage of SLMs for users lacking access to state-of-the-art hardware. Domains of interest include medical diagnosis in low-resource hospitals and offline decision support in remote environments, where deploying massive LLMs is infeasible, making it essential to support advanced reasoning on limited compute.

Our experiments show that TTA* consistently outperforms zero-shot chain-of-thought on multiple mathematical reasoning tasks, highlighting the effectiveness of combining classical search with modern LLMs. However, these results remain limited to mathematics, and the broader applicability of TTA* to areas such as commonsense reasoning, scientific discovery, or other complex tasks remains unexplored.

Future work should expand evaluation beyond mathematics and integrate TTA* into more practical environments to test real-world reasoning capabilities. Even so, our current version of TTA* represents a step toward making advanced reasoning accessible not only to researchers with large compute budgets, but also to real-world users operating under constrained conditions.

## ETHICS STATEMENT

This work focuses on enhancing the reasoning capabilities of small language models (SLMs) through a training-free, test-time scaling approach. Our method does not require new datasets, human annotation, or proprietary teacher models, thereby avoiding concerns related to data privacy, human subject research, or the reinforcement of biases through supervised fine-tuning.

However, language models can reflect and sometimes amplify biases present in their training data. While our approach improves reasoning performance, it does not explicitly mitigate these risks. Therefore, outputs should be carefully validated before deployment in high-stakes domains. Special caution is needed in sensitive applications such as education, healthcare, or decision-making, where incorrect outputs could have harmful consequences.

We emphasize that our framework is lightweight and deployable in resource-constrained settings, potentially broadening access to advanced reasoning capabilities. This democratization of model reasoning has positive implications for accessibility and inclusion. Nevertheless, transparency about model limitations, careful evaluation in the intended application domain, and safeguards against misuse remain essential.

Overall, we present our contribution as a technical step toward practical, efficient test-time reasoning for small models, while noting that it is not a substitute for responsible oversight in real-world applications.

## REPRODUCIBILITY STATEMENT

We aim to make our results fully reproducible. All code, exact prompts, and evaluation scripts used in this paper are provided in the project repository. We summarize the most important details so readers can re-run our experiments and verify our claims in Appendix A.

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

# A   APPENDIX

## A.1   EXPERIMENTAL CONFIGURATIONS

**Datasets and models.**   Our experiments used publicly available benchmarks: GSM8K, MATH500, MATH401, and the AIME (2024) problems. Models are used off-the-shelf; the exact model families and identifiers used in the paper are listed in the repository. Our code loaded models from the Hugging Face Hub.

**Hyperparameters**   The exact hyperparameters used for reported experiments are in the repository.

Key settings include:

- Decoding temperature: 0.3 (used in all reported runs).
- Children per expansion: 2 (two child nodes generated for each expanded node).
- Self-evaluation samples per node: 3 independent evaluations (scores averaged to compute the node reward).
- Early stop threshold for score: 95 (if any candidate reaches this score the search may terminate early).
- A* path-cost weight $w$: default value used is 3.
- Maximum iterations: defaulted value used is 8.

**Environment and dependencies.**   Experiments were run in a Python 3.10+ environment using PyTorch and Hugging Face `transformers`. For GPU-based reproduction we used standard CUDA-capable drivers.

**Evaluation and scoring.**   To maximize the reliability of our experimental results, we used GPT-4 as our "grader." After each experiment, GPT-4 computed accuracy on each benchmark using exact-match scoring rules.

**Licensing and model access.**   Code in the repository is released under the MIT license.

## A.2   PROMPTS

---

**Prompt for Critique**

Question: question
Answer: answer
Please provide detailed constructive criticism, yet highlight what is already correct.
Point the student in the right direction. Do not solve the problem.
Provide a grade (out of 100) in the format 'Grade: xx'.

---

**Prompt for Generating Child Nodes**

Question: question
Previous Answer: previous answer
Critique: critique
Given the feedback above, please try to solve the original problem again, step by step.

---

