# OpenReview forum: "Test-Time Scaling for Multistep Reasoning in Small Language Models via A* Search"
_ICLR.cc/2026/Conference — Submitted to ICLR 2026_

### Official Review · Reviewer_3CwX · 2025-10-26

**Soundness:** 2
**Presentation:** 2
**Contribution:** 2
**Rating:** 2
**Confidence:** 3

**Summary:**

This paper proposes TTA* (Test-Time A* Search) to equip small language models (SLMs) with structured reasoning capabilities through A*-based tree search. TTA* is claimed to be practical in resource-constrained settings due to the fact that not training or external supervision is required. Experimental evaluations indicate that models using TTA* yield improved performance on math reasoning tasks relative to a baseline where TTA* is not used.

**Strengths:**

(+) The proposed approach integrates merits of test-time scaling where (small) models improve answers via structured search at inference time and self-reflection where intermediate outputs are evaluated by the model. At the same time shortcomings of the former in terms of challenges in deployment and the latter in terms of increasing errors in multi-step tasks are overcome by the proposed TTA* methodology.

(+) The lack of necessity for training or external supervision promises to be greatly beneficial in accelerating deployment at scale.

(+) Models considered for experiments are open-source and chosen to reflect use of LLMs/ SLMs in resource-constrained environments with commodity hardware. Experimental results demonstrate merits of the proposed approach in terms of better accuracy and robustness on math reasoning tasks relative to a baseline that does not use TTA*.

**Weaknesses:**

(-) The logical flow of Fig. 1 is not clear. Specifically, how are the quantities $f(A), f(B)$ computed and what is the value of terms that make up $f(\cdot)$ for this example? While some details are specified in Eqn. (2), the paper will benefit from a more detailed caption for this figure.

(-) With reference to Sec. 3.1, the representation of the LLM reasoning process as a tree whose depth might be large could contradict with the inherent feature of SLMs lacking the same level of depth as LLMs. Some insight on this aspect related to how the tree-depth is managed by TTA* will be helpful.

(-) The discussion on Related Work spans 1.5 pages in the main paper. I wonder if a large part of this section can be moved to an Appendix, and the resulting space used to present more detailed experimental/ analytical results.

(-) In the context of related work, how does the approach of this paper compare with the treatment in [1, 2]?

[1] Li et al., Small Models Struggle to Learn from Strong Reasoners, ACL Findings, 2025.

[2] Zhang et al., Making Small Language Models Efficient Reasoners: Intervention, Supervision, Reinforcement, ICML 2025 Workshops.

(-) Experimental evaluations on other widely used math benchmarks including OlympiadBench [3] might be helpful to demonstrate the performance of the proposed TTA*.

[3] He et al., OlympiadBench: A Challenging Benchmark for Promoting AGI with Olympiad-Level Bilingual Multimodal Scientific Problems, ACL 2024.

(-) Perhaps the most significant shortcoming of the paper is that the experiments do not provide a point of comparison with other approaches that have been recently used to evaluate SLMs on math reasoning tasks. Demonstrating that reasoning accuracy with TTA* is higher across a range of setups than without TTA* is reasonable to expect, and does not provide fine-grained insights into the effectiveness of TTA*. The topic and domain examined by the paper are very interesting; I wonder if the paper will benefit from additional time to present more thorough experimental evaluations and comparisons with current art.

**Questions:**

Please see Weaknesses, above.

---

### Official Review · Reviewer_gx6t · 2025-10-27

**Soundness:** 2
**Presentation:** 2
**Contribution:** 3
**Rating:** 2
**Confidence:** 3

**Summary:**

This paper proposes a test-time scaling framework called Test-time A* Search (TTA*) that enhances the reasoning ability of Small Language Models (SLMs) by recasting reasoning as a search over a tree of partial solutions, without requiring additional training, external supervision, or multi-model orchestration.

The main contributions include: formulating reasoning as a guided tree search, designing an A*-style cost function to guide the search, and conducting comprehensive experiments across multiple reasoning benchmarks and SLM backbones.

**Strengths:**

The proposed idea is simple yet effective, offering several clear advantages: it requires no additional training, no external supervision, and no orchestration of multiple models. Despite its simplicity, the framework demonstrates substantial improvements in the reasoning capabilities of Small Language Models (SLMs) across diverse benchmarks.

Importantly, the paper introduces a novel adaptation of A* search for tree-based reasoning, a direction that, to the best of our knowledge, has not been systematically explored before. I think this direction is intuitively worth exploring.

**Weaknesses:**

The paper feels like it was put together a bit in a rush.
1. There’s not much analysis beyond the basic results shown in Table 1 and Figure 2.
2. Some important details are missing. For example, it’s not clear how they estimate the to-go cost, and the code repo looks unfinished.
3. There are quite a few small mistakes, like using \citet instead of \citep, and even a duplicated section title (4.3).
Overall, it’s a neat idea and definitely worth exploring, but the paper just isn’t quite ready for publication yet.

**Questions:**

1. How exactly is h(n) defined. I know it depends on the Reward function but I also haven't seen its definition either. Have you tried to vary the definition of g(n) and h(n) and how does that impact the final performance? How are the results affected by hyperparameters? Why using the magic number 100 in h(n)? I think there are lots of ablation study to be done regarding the cost functions.
2. You mentioned that A*-style cost function mitigate hallucinations and compounding errors, can you elaborate on this. What is the fundamental reason that you think A* search can solve this issue?

---

### Official Review · Reviewer_4k2h · 2025-10-30

**Soundness:** 2
**Presentation:** 3
**Contribution:** 2
**Rating:** 2
**Confidence:** 4

**Summary:**

This paper proposes Test-time A* (TTA*), a training-free test-time search procedure for small language models (SLMs). The method treats multi-step reasoning as a tree search: each node is a candidate solution, edges are revisions, and node expansion is prioritized using an A*-style score f(n)=g(n)+h(n)f(n)=g(n)+h(n)f(n)=g(n)+h(n), where g(n)g(n)g(n) is a depth penalty and h(n)h(n)h(n) is the model’s own self-evaluation score. The system iteratively critiques and revises its answers, generates two child candidates per step, and stops early if a high self-score is reached. Experiments on GSM8K, MATH500, AIME (2024), and MATH401 show accuracy gains over a zero-shot chain-of-thought baseline for several LLaMA and Qwen variants.

**Strengths:**

1) The focus on SLMs and test-time methods is well motivated. Improving multi-step mathematical reasoning in small models under deployment constraints (limited VRAM, edge/low-resource hardware) is important.
2) The method is simple and self-contained. It does not require fine-tuning, verifier models, or orchestration with a larger model.
3) The reported gains are consistent across multiple benchmarks and model families.

**Weaknesses:**

1) Unfair comparison. Results are only against zero-shot CoT baselines. Since TTA* is a multi-inference search, it should be compared with other test-time scaling or search-based methods (e.g., MCTS, self-refine, majority-vote). The reported gains therefore overstate relative performance.


2) No compute or latency analysis. Despite claiming efficiency for resource-limited settings, the paper reports accuracy only. Each iteration involves multiple generations, critiques, and self-evaluations, yet runtime, token, and cost comparisons are missing.


3) Missing sensitivity/ablations. Analysis of performance as a function of: number of iterations, children per expansion, number of self-evaluation samples, early-stop threshold, the weight in depth penalty, and temperature is not provided. These are important factors governing the accuracy–cost trade-off.

4) Table 2 is misleading for practicality. The parameters/VRAM/accuracy presentation is not tied to specific datasets, and includes speculative figures (e.g., for closed models).

5) Limited novelty. The core loop—generate, self-critique, and refine—is a standard pipeline in self-reflection work. The main addition of ranking by a depth penalty needs ablations demonstrating that it materially contributes to the gains.

**Questions:**

1) Why was GPT-4, which could introduce avoidable variance, used as the grader instead of exact-match deterministic scoring?
2) ​​What is the runtime and token usage overhead of TTA* compared to the zero-shot baseline?
3) Have you evaluated how accuracy changes with respect to the number of iterations, children per expansion, or self-evaluation samples?
4) How sensitive are the reported improvements to sampling randomness (the inference temperature of 0.3)?

---

### Official Review · Reviewer_4Pv2 · 2025-10-31

**Soundness:** 2
**Presentation:** 3
**Contribution:** 2
**Rating:** 4
**Confidence:** 3

**Summary:**

This paper explores test-time scaling with TTA*, a training-free, single-model test-time search framework that wraps a small language model (SLM) with an A*-guided exploration over a tree of partial solutions. Each node in the tree is a set of partial answer, critique, and score of the partial answer. TTA* starts with a single prompt, then generate answer, critique, and score all by the same single model. Then expand each current leaf with two nodes until a maximum iteration or a threshold is reached.
The paper evaluates A* for LLM on GSM8K, MATH500, AIME-2024, MATH401 with models from LLaMA and Qwen

**Strengths:**

1. Simplicity: the compute cost of small model without retraining is low in expectation, making it easier to implement. Also the single-model regime and the principled search are both well motivated and simple.
2. Provided pseudo code for simple implementation
3. Consistent and broad gains across the benchmark suite

**Weaknesses:**

1. Baselines and compute parity. The paper compares against zero-shot CoT only. For a test-time scaling paper, omitting other baselines makes it hard to attribute gains to A* rather than simply “more tries + critique.” Compute-matched comparisons (tokens/runtime) are needed. In related works a lot of test-time scaling methods are surveyed but not evaluated against. I know they have some discrepancies in terms of setup, but a best-effort try and better notation should be included



Minor:
1. Typo at the end of line 188
2. line 377: "All baselines are evaluated against zero-shot chain-of-thought Wei et al. (2023), and full prompts are provided in the appendix." aren't baselines just zero-shot cot? So it should be all baselines are zero-shot cot.

**Questions:**

1. Clarity: at the end of section 3, "The final answer is selected as the candidate with the highest
self-evaluation score among all generated nodes, ensuring both efficiency and high-quality solutions.", is self-evaluation score `f(n)` or just `h(n)`?

---

### Official Review · Reviewer_tNUi · 2025-10-31

**Soundness:** 1
**Presentation:** 1
**Contribution:** 1
**Rating:** 2
**Confidence:** 5

**Summary:**

This paper proposed Test-Time A* Search, a training free framework that applies A* search to improve mathematical reasoning in small language models. The method constructs a tree of candidate solutions, using the model's self evaluation as a heuristic to guide search, and evaluate TTA* on mathematical benchmarks using SLM.

While the paper claims consistent improvements, the work suffers from fundamental methodological flaws, inadequate experimental validation, and poor presentation quality that render it unsuitable for publication.

**Strengths:**

1. Relevant Problem: Addressing reasoning capabilities and test-time scaling methods in resource-constrained SLM is an important practical problem.
2. Training-free Method: The method does not require additional fine tuning or external reward models, which could be beneficial for deployment scenarios.

**Weaknesses:**

1. Unreliable Self-Critique Foundation: The entire framework relies on SLM’s self-evaluation capabilities to compute the heuristic function h(n). However, the authors themselves acknowledge that self-reflection in SLMs is noisy and unreliable and that SLMs “can be noisy, leading to overconfidence in incorrect answers or excessive critique in accurate responses”(L-142). This fundamentally undermines the core premise of using self-evaluation scores in the A* cost function. The paper provides no empirical validation of the reliability or calibration of these self-assigned scores, which is a critical omission given that the entire search process depends on them.
2. No Analysis of Hallucination Mitigation: The authors claim TTA* helps mitigate hallucinations and compounding errors, yet provide no evidence that their method actually reduces them. The self-critique mechanism could easily amplify rather than reduce hallucinations if the model confidently assigns high scores to incorrect solutions.
3. Missing Critical Baselines: The paper only compares TTA* against zero-shot CoT, which is wholly inadequate given the additional computational cost. The authors should compare with other baseline such as Best of N sampling; majority voting, and other test-time scaling methods, e.g. [1]
4. No Ablation Studies.The authors do not conduct any ablation regarding the method.
5. Poor writing quality and presentation: The overall writing is very poor, including redundant content; orphaned content; Duplicate Section Numbers and it is Verbose and Unfocused.

[1] Tree of Thoughts: Deliberate Problem Solving with Large Language Models

**Questions:**

1. Can you provide empirical evidence showing the correlation between self-assigned scores and actual correctness? What is the calibration error of these self-evaluations across different model sizes and problem difficulties?
2. The authors claim to mitigate hallucinations, but provide no measurements.
3. How does TTA* compare to Best-of-N sampling or self-consistency when given the same total number of forward passes?
4. Why did you only compare against zero-shot CoT when the literature contains numerous stronger baselines (self-consistency, Best-of-N, other test-time scaling methods)?
5. What is the purpose of Table 2, and why compare TTA*-enhanced 8B models to non-TTA* 70B models? This seems designed to create a misleading impression.

---

### Meta-Review · Area_Chair_8T9C · 2025-12-29

**Summary:**

This paper proposes TTA*, a training-free test-time A* search framework to improve multi-step reasoning in small language models. The idea of structured test-time search is interesting and the problem setting is relevant.

However, the paper has significant shortcomings. Most critically, the evaluation is weak: results are only compared against zero-shot chain-of-thought, with no compute-matched or stronger test-time scaling baselines (e.g., Best-of-N, self-consistency, or other search-based methods), making the claimed gains difficult to attribute to the proposed method. The approach also relies heavily on self-evaluation scores as the search heuristic, yet provides no validation of their reliability or calibration, which undermines the core methodology. In addition, there is no analysis of computational cost, sensitivity, or ablations, and the presentation quality has multiple issues. I therefore recommend rejection.

**Reviewer Concerns:**

The authors do not provide a rebuttal; hence, all concerns are unresolved.

**Reviewer Scores:**

The scores after rebuttal: 2,2,4,2,2

---

### Decision · Program_Chairs · 2026-01-26

Reject